# Breakfast Consumption Suppresses Appetite but Does Not Increase Daily Energy Intake or Physical Activity Energy Expenditure When Compared with Breakfast Omission in Adolescent Girls Who Habitually Skip Breakfast: A 7-Day Randomised Crossover Trial

**DOI:** 10.3390/nu13124261

**Published:** 2021-11-26

**Authors:** Julia Kirstey Zakrzewski-Fruer, Claire Seall, Keith Tolfrey

**Affiliations:** 1Institute for Sport and Physical Activity Research, School of Sport Science and Physical Activity, University of Bedfordshire, Bedford MK41 9EA, UK; claire.seall@bps.org.uk; 2Paediatric Exercise Physiology Research Group, School of Sport, Exercise and Health Sciences, Loughborough University, Loughborough LE11 3TU, UK; K.Tolfrey@lboro.ac.uk

**Keywords:** adolescents, children, exercise, health, nutrient timing

## Abstract

With concerns that adolescent girls often skip breakfast, this study compared the effects of breakfast consumption versus breakfast omission on free-living physical activity (PA) energy expenditure (PAEE) and dietary intakes among adolescent girls classified as habitual breakfast skippers. The participants went through two 7-day conditions in a trial with a crossover design: daily standardised breakfast consumption (energy content: 25% of resting metabolic rate) before 09:00 (BC) and daily breakfast omission (no energy-providing nutrients consumed) until 10:30 (BO). Free-living PAEE, dietary intakes, and perceived appetite, tiredness, and energy levels were assessed. Analyses were linear mixed models. Breakfast manipulation did not affect PAEE or PA duration. Daily fibre intake was higher (*p* = 0.005; d = 1.31), daily protein intake tended to be higher (*p* = 0.092; d = 0.54), post-10:30 carbohydrate intake tended to be lower (*p* = 0.096; d = 0.41), and pre-10:30 hunger and fullness were lower and higher, respectively (*p* ≤ 0.065; d = 0.33–1.01), in BC versus BO. No other between-condition differences were found. Breakfast-skipping adolescent girls do not compensate for an imbalance in energy intake caused by breakfast consumption versus omission through subsequent changes in PAEE but may increase their carbohydrate intakes later in the day to partially compensate for breakfast omission. Furthermore, breakfast can make substantial contributions to daily fibre intake among adolescent girls.

## 1. Introduction

Observational reports in children and adolescents suggest that more frequent breakfast consumption is associated with higher physical activity (PA) levels, which may contribute to reduced adiposity and cardiometabolic disease risk [1,2,3]. Indeed, a sustained energy deficit through increased physical activity energy expenditure (PAEE; i.e., bodily movements resulting in energy expenditures exceeding 1.5 metabolic equivalents (METs)) and/or reduced energy intake can reduce adiposity, and PA independently improves cardiometabolic health [4,5]. As such, the combination of consuming breakfast with the possible consequential improvements in PA could enhance an individual’s health. However, only experimental study designs can determine whether breakfast consumption causes such improvements when compared with breakfast omission [1,6]. Randomised controlled trials in adults suggest that rather than reductions in daily energy intake [7], increased free-living PAEE occurs in response to breakfast consumption (typically carbohydrate-based) versus breakfast omission, possibly due to exogenous glucose availability being the primary fuel source for PA [8,9,10]. So far, evidence of a breakfast effect on PA has been inconclusive and has not necessarily targeted the most relevant populations [11,12,13]. Understanding such responses in adolescent girls has particular public health relevance because the adolescent decline in breakfast consumption [1] and PA [14] is more pronounced in girls than in boys, such that only ~20% of UK adolescent girls consume breakfast daily [2], and only ~9% meet the PA level recommendations [15].

Adolescent girls may respond differently to breakfast manipulations than adults because they have distinct hormonal, metabolic, and behavioural profiles [16], including pubertal insulin resistance [17], growth [18], fuel utilisation [19,20], and PA behaviours [15]. Furthermore, eating breakfast to aid the provision of a continuous carbohydrate supply may be more important to fuel daily PA in adolescents than adults due to their higher reliance on exogenous carbohydrate [19,20] and higher energy expenditures [21]. Nevertheless, protein-rich breakfasts may be more beneficial when the primary aim is to reduce appetite [22,23,24]. In young people, only two published experimental studies have examined the causal effects of breakfast manipulation on PA [25,26], and cross-sectional findings are unclear [2,3,27,28], as are experimental findings on energy intake [22,23,24,25,29]. In our 3-day crossover trial comparing breakfast omission versus consumption in adolescent girls, PA assessed via wrist-worn accelerometry was unaffected despite the incomplete (24%) energy intake compensation after breakfast omission [25]. Using combined heart rate and accelerometry devices to provide a more sensitive measure of PAEE, our follow-up 7-day crossover trial reported that adolescent girls spent more time in light PA before 10:30 and after school, and less time sedentary after school during daily versus intermittent breakfast consumption [26]. However, PAEE was unaffected, and energy intake was not assessed. It is logical that contrasting the extremes of complete daily breakfast omission across the week with breakfast consumption should result in more pronounced effects, which can be ethically achieved with girls who habitually skip breakfast. Moreover, data from girls who habitually skip breakfast ensure that the findings can be applied directly to those ‘in need’. Thus, the primary aim of this cross-over study was to compare the effects of seven days of breakfast consumption with breakfast omission on free-living PAEE in adolescent girls classified as habitual breakfast skippers. The secondary aims were to examine the effects on dietary intakes and perceived appetite, energy, and tiredness. The key novel contributions of this study in extending previous breakfast–PA research were: (1) the targeting of adolescent girls who habitually skip breakfast specifically, (2) an extended breakfast omission experimental period of an entire week, and (3) an assessment of both PA and diet-related responses as outcomes.

## 2. Materials and Methods

### 2.1. Participants

This study was conducted in accordance with the ethical standards of the University of Bedfordshire Research Ethics Committee (ethical approval number: 2018SSPA002) and the Helsinki Declaration of 1975 as revised in 1983. The study was registered at clinicaltrials.gov with identifier NCT04481776. Data collection was completed between January 2018 and February 2019. Thirty-nine girls aged 11–14 years were recruited from schools located in Bedford, England. Parental informed consent and child assent were provided for all participants. Girls were excluded from the study if they had health-related issues identified from a health screening questionnaire (e.g., allergies to the breakfast meals, fitted with a pacemaker), were unable to walk or wear a combined heart rate and accelerometer on their chest, or were classified as habitual breakfast consumers during preliminary measures.

### 2.2. Sample Size Calculations 

The sample size estimation was based on our primary outcome, PAEE. A positive energy balance of at least 628 kJ/d in excess of normal growth requirements [30] may explain the higher adiposity in infrequent breakfast consumers [1]. Post-breakfast energy intake compensation was expected to account for ~24% of breakfast energy intake in adolescent girls [25], which equates to ~312 kJ of the expected average breakfast of 1302 kJ (i.e., 25% of the resting metabolic rate (RMR) based on an RMR of 5207 kJ/d) used here. Therefore, energy intake was expected to be 989 kJ/d higher on the breakfast consumption days as compared with the breakfast omission days in the present study (i.e., the added energy consumed at breakfast minus the 24% energy intake compensation). Based on these figures, we deemed that the smallest worthwhile difference in the estimated PAEE between the conditions would be 1617 kJ/d (i.e., 989 kJ/d to achieve energy balance plus 628 kJ/d [30]). The expected SD for free-living PAEE in adolescents is ~1990 kJ/d [26,31,32]. Thus, a sample size of 15 participants was estimated to detect a significant difference in the estimated PAEE at 85% power, with an α of 0.05, a Cohen’s f effect size of 0.43, and an assumed correlation between treatments of 0.5 in this two-treatment crossover design. Thirty-nine girls were recruited to allow for ineligible volunteers and dropouts.

### 2.3. Preliminary Measurements

Stature was measured to the nearest 0.01 m using a portable Leicester height measure (SECA Corporation, Hamburg, Germany). Body mass was measured, and percent body fat was estimated to the nearest 0.1 kg and 0.1%, respectively, using a Tanita Body Composition Analyser (BC-418 MA, Tanita Corporation, Tokyo, Japan). Body mass index (BMI) was calculated as body mass divided by stature squared (kg∙m^−2^), with weight status subsequently defined according to the International Obesity Task Force age- and sex- specific cut-points [33]. Waist circumference was measured to the nearest millimetre in accordance with recommended procedures [34]. With the assistance of a primary home-based carer, the girls provided a validated [35,36] self-assessment of their physical maturation using secondary sexual characteristics [37]. A questionnaire was used to assess their breakfast habits on week and weekend days, including frequency (number of days per week), time and location of consumption, types of food and beverages consumed, and reasons for skipping breakfast. For study eligibility purposes, the girls were asked the following question: ‘How often do you normally consume less than 50 kcal (e.g., less than a piece of fruit or a small glass of juice) before 10:30?’; the girls were given the opportunity to ask questions to clarify any aspects of this question that they did not understand and were asked to confirm their response verbally prior to the commencement of the study. Only girls who skipped breakfast (less than 50 kcal before 10:30) on at least four days/week were included in the study [8,9].

The participants then completed two tests required for the individual calibration of the combined heart rate and accelerometry devices used in the experimental conditions: (1) a submaximal treadmill exercise protocol consisting of 4 × 4 min stages to determine the relation between heart rate and estimated energy expenditure, and (2) a resting metabolic rate protocol where a 10 min resting expired air sample was collected after 20 min of quiet rest in the fasted state [38]. Expired air was sampled continuously during the treadmill and RMR tests using an online gas analysis (Metalyzer 3b, Cortex, Leipzig, Germany). Energy expenditure was estimated using the Weir equation [39,40].

### 2.4. Experimental Design

Using a cross-over design, each participant went through two 7-day conditions separated by a seven to ten-day washout: breakfast consumption (BC) and breakfast omission (BO). The conditions were completed in a counter-balanced order using block randomisation, with as close as possible to half of the participants randomised to each sequence (i.e., BC then BO or BO then BC), as determined using a computer-based number generator. The irregular menstrual cycles in this population coupled with the 7-day duration of each intervention, and the feasibility of allowing the girls to complete the study alongside a friend to help provide an enjoyable, exciting, and comfortable experience meant that it was not possible to align the experimental conditions to a specific menstrual cycle phase for each individual. On each of the seven days, the participants were asked to consume a standardised breakfast before 09:00 in BC and to abstain from all energy-providing nutrients before 10:30 in BO. These cut-off times were in line with proposed definitions of breakfast, which should be consumed within 2 to 3 h of waking, typically no later than 10:00 [41,42]. Due to the nature of the sample, it was also important that the breakfast omission cut-off time coincided with the participants’ first opportunity to consume food or drink at school (i.e., break time at ~10:30). The participants could eat as and when they pleased from 10:30 onwards in both conditions. Throughout each 7-day condition, free-living PAEE was estimated using the combined heart rate and accelerometry devices (Actiheart, CamNtech, Cambridge, UK), as described elsewhere [22], and dietary intakes were recorded using a combined photographic and written food diary, also described elsewhere [21]. Additionally, the participants completed a Visual Analogue Scale (VAS) for hunger, fullness, tiredness, and energy levels on the dietary assessment days. The experimental conditions did not coincide with anticipated changes in PA or dietary habits (e.g., holidays, school sports days), as confirmed with the participants, their parents, and teachers.

The participants and their parents received telephone reminders during each condition to help maximise compliance to study procedures (i.e., adhering to the breakfast omission and omission protocols, wearing the Actiheart monitor, and recording dietary intakes). Compliance to the breakfast intervention was confirmed via photographs taken by the participants of all food and drink consumed before 10:30 using the digital camera provided (ViviCam 46, Vivitar Shenzhen, China), and through a written daily breakfast log that included the time breakfast was consumed for BC or the time that the first meal or snack was consumed after the breakfast omission cut-off (i.e., 10:30) for BO. 

### 2.5. Breakfast Interventions

The quantity, composition, and time of consumption of the standardised breakfast was designed to align with proposed definitions of ‘breakfast’ [41,42]. The energy content was 25% of individual RMR, which is in line with recommendations that breakfast should contribute to ~15–25% of daily energy intake [41]. Based on our previous studies, RMR ranged from around 3348 to 7115 kJ/d in adolescent girls [26], which equated to a breakfast energy content of 837–1779 kJ/d in the present study. Prior to the experimental conditions, the participants selected one wholegrain, high-fibre, ready-to-eat cereal (with the option of adding raisins) and fruit juice from a limited selection that was based on the breakfast preferences of girls in our previous work [25,26]. The breakfast items that the girls chose are shown in Appendix A. The breakfast items selected were consumed on each day of BC; thus, breakfast composition was controlled within participants, but not between participants, in order to account for individual preferences. This CHO-based breakfast was chosen as higher exogenous glucose availability may act as a physiological mechanism that increases PA in response to breakfast consumption versus omission [8,9,10]. Furthermore, high fibre, wholegrain, cereal-based breakfasts may be particularly beneficial to health in adolescents [43,44]. The participants were instructed to consume the breakfast at home before 09:00. To ensure that the correct amount of each breakfast item was consumed, the food items were provided to the participants in pre-packaged containers, and the participants were provided with a marked beaker to measure their milk and juice each morning. The only exception was that parents were asked to provide the semi-skimmed (1.8%) milk. To be included in the final dataset, the participants were required to confirm verbally and using photographic evidence that they had consumed their breakfast on all seven days of BC, and to confirm verbally that they had consumed no energy-providing nutrients before 10:30 on all seven days of BO.

### 2.6. Physical Activity Energy Expenditure Assessment

Participants were fitted with a combined heart rate and accelerometer (Actiheart, CamNtech, Cambridge, UK) the day before each condition, which was removed after eight days. Combining heart rate and accelerometer data improves the validity of PAEE estimations in 12–13-year-olds as compared with accelerometry or heart rate monitoring alone [45]. The procedures used to fit the monitor and the instructions provided to the participants to ensure that only genuinely meaningful behavioural responses were recorded were in line with recommendations and are described in detail elsewhere [26]. Each participant’s monitor was set to record data in 15-second epochs and was individually calibrated using the measured RMR and exercise energy expenditure values from preliminary testing. This calibration method accounted for individual differences in the heart rate–PAEE relationship, ensuring greater accuracy of the PAEE estimations when compared with group regression equations [45].

As only Actiheart data during waking hours were analysed, participants were required to record their answers to the following questions using a daily log: “what time did you wake up?”, “what time did you get out of bed?”, “what time did you turn off the light and go to bed?”, and “what time did you fall asleep?”. Using a standardised protocol, the self-reported wake and bedtimes were utilized to provide a region of interest for each 24 h Actiheart data file. Objective markers were then used to identify bed time (i.e., the beginning of prolonged minimal movement accompanied by a decline in heart rate) and wake time (i.e., the beginning of prolonged increased movement accompanied by an increase in heart rate) [26,32].

The procedures used to analyse the Actiheart data files are described in detail in our previous publication [26]. After excluding data classified as ‘lost’ and ‘not worn’, only datasets with four valid days (i.e., at least 10 h of useable data), including one weekend day for each condition, were included [2,26,28]. Branched equation modelling was used to estimate PAEE. Metabolic equivalent (MET) values were used to define sedentary (<1.5 METs), light (1.5–2.9 METs), moderate (3.0–5.9 METs), and vigorous (>5.9 METs) activity.

### 2.7. Dietary Assessment

Weighed food records were not considered to be suitable for the present study due to the high participant burden and poor compliance in adolescents [46,47], a population that has reported a preference for methods using technology such as a disposable camera [48]. Thus, the participants recorded their daily diet using a digital camera (ViviCam 46; Vivitar) and a written food diary during the final four consecutive days that included two weekdays and both weekend days during each condition. As dietary intakes vary significantly between Saturday and Sunday, including both weekend days and a selection of weekdays was recommended in order to obtain estimates that are representative of usual intakes [49]. Thus, all four days had to have been completed for a participant to be included in the final dataset for dietary analyses. The combined photographic and written food diary is described in detail elsewhere [25]. We have previously shown that the natural variation in free-living energy intake assessed using this method may be small enough to detect meaningful differences [25]. The mass of each food and beverage item consumed was estimated by comparing the digital photographs taken by the participants with the Young Person’s Food Atlas [50,51], which has good agreement with weighed food diaries in children aged ≥11 years [52]. Energy and macronutrient intakes were estimated from the food diaries using the myfood24 online dietary analysis software (Nexus, Leeds, UK).

### 2.8. Perceived Appetite, Tiredness, and Energy Levels

Participants were asked to answer the following questions using a 100 mm visual analogue scale (VAS): “How hungry do you feel right now?”, “How full do you feel right now?”, “How tired or drowsy do you feel right now?”, and “How energetic do you feel right now?”. Responses were recorded on the four days that dietary intakes were assessed, at three time points: on waking (i.e., ‘baseline’), at 09:00 (i.e., the BC cut-off), and at 10:30 (i.e., the BO cut-off); further assessments throughout the day were not taken due to the additional participant burden. VAS has been shown to be valid and reliable for assessing hunger, fullness, fatigue, and energy level in adults [53,54,55], and it has been successfully used in adolescent girls to assess perceptions of appetite and mood in free-living settings [56]. Based on the number of complete VAS available and the previous literature [21,56], participants were required to have had at least three days of complete VAS data per condition in order to be included in the final dataset for VAS analyses; the mean of the three days was calculated.

### 2.9. Statistical Analyses

Statistical analyses were completed using the IBM SPSS statistics software for Windows version 26 (IBM Corporation, New York, NY, USA). One-way ANOVA was used to test for differences in participant characteristics and the nutrient content of the breakfast meal by outcome-specific analytical samples. As breakfast manipulation may affect PA and diet during specific times of the day [2,8,9,10,26], estimated PA and dietary variables were computed for three daily time segments: from wake time to before 10:30, from 10:30 to before 15:30, and from 15:30 until bed time [26]. These times coincided with the 10:30 breakfast omission cut-off and the end of the school day to account for potential differences in PA during and outside of school time [57]. The Shapiro–Wilks tests showed that the residuals were not normally distributed for the time spent in moderate and vigorous PA, for the PAEE from sedentary, light, moderate, and vigorous activity and total PAEE, and for post-10:30 CHO and fibre intakes (*p* ≤ 0.05). Thus, for consistency, all PA and dietary outcomes were natural log-transformed (Ln) and presented as a geometric mean (95% confidence intervals (CI)), with analyses based on ratios of the geometric means and a 95% CI for the ratios. Linear mixed models were used to examine all outcome variables, with condition and time of day included as fixed factors. The linear mixed models included a random effect for each participant and were adjusted for period (order) effects [58]. Minutes of useable data (Ln) was included as a covariate for PA-related variables. Where significant condition or condition by time-of-day interactions were found, post hoc analysis was performed using the Holm–Bonferroni correction for multiple comparisons; data from each individual time segment were compared between the conditions for significant condition by time interactions [59]. Statistical significance was accepted as *p* ≤ 0.05; *p* values with two decimal places are attributed to non-significant results or three to at least borderline significant results. Pooled *p* values are provided in the text where appropriate; the 95% CI for the differences are provided for all primary and secondary outcome variables in the tables. Absolute standardised effect sizes (Cohen’s d) are provided to supplement important findings (i.e., potentially meaningful between-condition differences), with 0.2 considered the minimum important difference, 0.5 moderate, and 0.8 large [60]. Values are presented as means ± SDs for descriptive data; for the results of statistical analyses, values are presented as estimated marginal means or geometric means (95% confidence intervals (CIs)) unless stated otherwise. Figures of individual responses are provided in order to give a more detailed insight into inter-individual variability where appropriate.

## 3. Results

### 3.1. Participant Characteristics

The final sample included 15 participants for PA analyses, 11 participants for dietary analyses, and 11 participants for VAS analyses; the flow of participants from enrolment to analyses is shown in Figure 1. The physical characteristics of the participants are shown in Table 1. There were no significant differences in the physical characteristics or breakfast frequencies between the final samples for each outcome (*p* ≥ 0.85 for all). 

### 3.2. Breakfast Meals

The standardised breakfast energy and macronutrient intakes of the girls who were included in the PA, dietary, and VAS analyses are shown in Table 2; there were no significant differences in these variables between the samples (*p* ≥ 0.43). Data from the food diaries indicated that on average, the girls consumed their first meal or snack at 11:48 ± 00:52 in BO.

### 3.3. Wake Time and Useable Data 

Useable data averaged at 910 min/d for BC and 942 min/d for BO. Minutes of useable data (Ln) were lower in BC versus BO across the three time segments (BC 305 (297–313) vs. BO 316 (308–324) min/d; condition main effect *p* = 0.017; condition by time-of-day interaction *p* = 0.20). For descriptive purposes, mean (SD) wake time did not differ significantly between the conditions (BC 07:06 ± 00:17 vs. BO 07:00 ± 00:22; *p* = 0.26), whereas bedtime was earlier in BC versus BO (BC 23:00 ± 00:32 vs. BO 23:13 ± 00:26; *p* = 0.042).

### 3.4. Physical Activity Energy Expenditure and Duration 

Table 3 shows the estimated daily PAEE and the time spent in PA for each intensity, stratified by condition and time of day. Data for all PAEE variables and minutes of useable data were natural log-transformed. Adjusting for minutes of useable data, the main effects for the condition and the condition by time-of-day interaction were non-significant for the estimated PAEE from sedentary, light, and moderate activities (*p* ≥ 0.41 for all). Total PAEE and vigorous PAEE both tended to be lower in BC versus BO, but the effect sizes were trivial-small (*p* ≤ 0.097; d = 0.18 to 0.20) and the condition by time interactions were non-significant (*p* ≥ 0.21 for all). Individual responses for total daily PAEE are shown in Figure 2. In 10 out of the 15 participants, PAEE was lower in BC than BO, and the difference exceeded 628 kJ/d in four girls. Consequently, for the remaining five girls, PAEE was higher in BC than in BO, and this difference exceeded 628 kJ/d for one of them. The main effect of time of day was significant for the PAEE for each intensity and for the total PAEE (*p* < 0.0005 for all); after adjusting for multiple comparisons, energy expenditure from sedentary activities tended to be lower at wake–10:30 as compared with 10:30–15:30 (*p* = 0.083), and light PAEE tended to be higher at 10:30–15:30 versus 15:30–bed (*p* = 0.085).

Data for the time spent sedentary and in light, moderate, and vigorous PA were log-transformed. Adjusting for minutes of useable data, sedentary time tended to be higher in BC versus BO, although the effect size was trivial (*p* = 0.068; d = 0.13), and the condition by time interaction was not significant (*p* = 0.36). The main effects of condition and the condition by time-of-day interaction were not significant for the time spent in light or moderate PA (*p* ≥ 0.34 for all). Time spent in vigorous PA tended to be lower in BC versus BO, and the condition by time interaction tended to be significant (*p* ≤ 0.077), but again, the between-condition effect size was trivial (d = 0.18). The main effect of the time of day was significant for all PA intensities (*p* ≤ 0.001 for all), but there were no significant differences between the individual time segments after adjusting for multiple comparisons (*p* ≥ 0.14 for all). 

### 3.5. Energy and Macronutrient Intakes

Table 4 shows the total and post-10:30 daily energy and macronutrient intakes stratified according to condition. For total daily intakes, there were no significant effects of condition for energy, CHO, and fat (*p* ≤ 0.63 for all), whereas protein intakes tended to be higher (*p* = 0.10; d = 0.54) and fibre intakes were significantly higher (*p* = 0.01; d = 1.31) in BC versus BO. Given the seemingly very high accuracy of energy intake compensation at the group level shown in Table 4, Figure 3 shows the difference in total daily energy intake between the conditions at the individual level. Based on the 628 kJ/d cut-off (30), differences in total daily EI were accurate in four of the girls. There was one girl with a relatively extreme difference; removal of this outlier did not affect the statistical analyses but actually tightened the group means even further. The main effect of condition and condition by time interaction were non-significant for post-10:30 energy, fat, protein, and fibre (*p* ≥ 0.12 for all). Post-10:30 CHO intake tended to be lower in BC versus BO (*p* = 0.096; d = 0.41), with no significant condition by time-of-day interaction (*p* = 0.50). Figure 4 shows individual responses for post-10:30 CHO intake; 8 of the 11 girls consumed less CHO after 10:30 in BC versus BO. The main effect of time was non-significant for all post-10:30 dietary variables (*p* ≤ 0.17 for all) other than fibre intakes, which were higher at 10:30–15:30 than at 15:30–bed (*p* = 0.048).

### 3.6. Perceptions of Appetite, Tiredness, and Energy Levels

Table 5 shows the perceived appetite, tiredness, and energy levels in the morning, stratified by condition. The main effect for condition approached significance for perceived hunger, which tended to be lower in BC versus BO, with a small effect size (*p* = 0.063; d = 0.33). There was a significant main effect of time (*p* = 0.003) and condition by time interaction (*p* = 0.001) for hunger; whilst there were no significant differences over time in BC (*p* ≥ 0.97 for all), hunger was lower at waking versus both 09:00 and 10:30 in BO (*p* ≤ 0.012; d = 0.87–1.54) and tended to be lower at 09:00 versus 10:30 (*p* = 0.074; d = 0.67). Perceived fullness was higher in BC versus BO (*p* = 0.045; d = 0.35); the condition by time interaction (*p* = 0.010) was significant, whereas the main effect of time of day was not (*p* = 0.23). Fullness was higher in BC versus BO at 10:30 (*p* = 0.004; d = 1.01) and approached significance at 09:00 (*p* = 0.065; d = 0.58). Perceived tiredness was not different between the conditions, and the condition by time interaction was also non-significant (*p* ≤ 0.71 for both); the main effect of time was significant (*p* < 0.0005), with tiredness being higher on waking versus 09:00 and 10:30 (*p* ≤ 0.006). Similarly, perceived energy level was not different between the conditions, and the condition by time interaction was also non-significant (*p* ≤ 0.62 for both); the main effect of time was significant (*p* < 0.0005), with energy levels being lower on waking versus 09:00 and 10:30, and lower at 09:00 versus 10:30 (*p* ≤ 0.017 for both).

## 4. Discussion

In adolescent girls who habitually skip breakfast, this 7-day crossover trial reported for the first time that PA duration and PAEE did not differ meaningfully between BC and BO. Total daily energy intake was almost identical between the conditions, which coincided with increased perceived morning appetite and a tendency for increased carbohydrate intake in response to BO (i.e., after 10:30). Thus, the girls were able to accurately compensate for the additional or missed energy at breakfast after 10:30.

Our finding that daily total PAEE and PA duration did not differ between BC as compared with BO agrees with some studies on adult samples, including female habitual breakfast skippers [12], obese mixed-sex habitual breakfast consumers and skippers [9], and mixed-sex habitual breakfast consumers and skippers varying in weight status [11,13], yet it contradicts research on adults showing that BC can increase PA in female habitual breakfast consumers [10] and lean mixed-sex habitual breakfast consumers and skippers [8]. The current findings also contrast with our previous crossover studies in adolescent girls, where the time spent in light PA was higher and after-school sedentary time was lower in daily BC versus intermittent BC and BO [26], although PA was unaffected by three days of BC versus BO when assessed using less sensitive measures [25]. It is possible that the girls in the current study responded differently because they had adapted metabolically and behaviourally to skipping breakfast habitually, whereas the girls in our previous research generally consumed breakfast habitually [25,26]. In the current sample, the 10:30 breakfast omission cut-off may have been too early to detrimentally affect PA, with studies on adults employing cut-offs at 12:00 [8,9,10,11] or 11:30 [12], which is difficult to justify ethically in adolescent girls. Additionally, MVPA averaged ~85 min/d in our sample, indicating high motivation for PA that may not have been overridden by breakfast manipulation. Interestingly, examination of the individual differences in total PAEE between the conditions provided insight into the high inter-individual variability in the responses to breakfast manipulation. Such inter- (and intra-) individual variability is common in the field of energy balance [61,62,63] and requires consideration in future breakfast-related work, along with addressing the potential modifying effect of breakfast and PA habits on the breakfast–PA relationship.

Under free-living conditions, low-intensity and sporadic PA may be most sensitive to the effects of breakfast consumption, whereas moderate or vigorous PA is typically planned and structured [8,26]. As such, the tendency for higher vigorous PAEE and total PAEE during BO is more likely to have resulted from a particularly active planned lesson at school or a sports club for some participants, for example, rather than whether or not they consumed breakfast. On this note, environmental constraints that are inherent to the lives of adolescent girls, including the school timetable, extra-curricular activities, family routines, and commitments, mean that it is difficult to truly assess ‘free-living’ PA responses in this population that would be amenable to any intervention effects. Thus, this may have contributed to the lack of difference in PA between the conditions, and future research may consider collecting data on such planned activities. It is also possible that the lower perceived energy levels and heightened tiredness early in the day across the conditions in our sample of female adolescent breakfast skippers may have overridden the potential effect of breakfast to increase morning PA, which is the time of day that is particularly sensitive to breakfast manipulation [8,9,10]. In support of this finding, adults whose activity peak is towards the later part of the day tend to skip breakfast [11]. As such, further research is warranted to examine the influence of chronotype on the breakfast–PA relationship in adolescent girls and to include measures of PA that may be particularly modifiable (e.g., PA during the school holidays or under laboratory conditions).

With total daily energy intake being almost identical between the conditions, post-10:30 energy intake was sufficient to compensate for the energy consumed or missed at breakfast. This finding is in accordance with previous research comparing carbohydrate-based BC versus BO in mixed-sex groups of so-called ‘adolescents’ who habitually skipped breakfast [22,24]. Nevertheless, the age range of 13–17 years [22] or the mean age of 19 years [24] in this past research indicates that the samples were either a mix of early- and late-pubertal adolescents [22] or a mix of late-pubertal adolescents and young adults [24]; furthermore, these findings were not consistent, with one study reporting increased daily energy intake during BC in ‘breakfast skipping’ overweight/obese females aged 15–20 years [23]. Moreover, total daily energy intake was higher during BC versus BO in adolescent girls [25] and in 8–10-year-old children [29] who tended to consume breakfast regularly. It is possible that the habit of skipping breakfast involves learning new eating patterns such that the habitual breakfast skippers in our study and previous work [22,24] were able to compensate by consuming extra energy later in the day. In adults, however, a recent systematic review of randomised controlled trials showed that the total daily energy intake was higher during periods of BC compared with BO regardless of breakfast habit [7]. The higher perceptions of hunger and lower fullness in response to breakfast omission in our study may have contributed to the accuracy of energy intake compensation between BC and BO, which complements previous findings on children [29] and adolescents classified as breakfast skippers [22] in controlled laboratory settings. However, no effect was found in women who habitually skipped breakfast, perhaps because they were older and thus more accustomed to skipping breakfast [12]. The energy intake compensation in response to BO was primarily in the form of carbohydrates, perhaps because the standardised breakfast was carbohydrate-rich, and dietary compensation may be macronutrient-specific [64], which aligns with previous research on adult women who habitually skipped breakfast [12]. In terms of diet quality, consuming breakfast also contributed to higher total daily fibre intakes, which can favourably affect long-term cardiometabolic health and thus has important health implications [43,44]. Overall, these findings warrant further research on the potential modifying effects of breakfast habits and age on the causal link between breakfast and dietary intakes, considering whether any compensatory behaviours are conscious and planned (i.e., to prepare for the next day) or unconscious and spontaneous.

Limitations of the present study include the 7-day intervention period; indeed, our sample of adolescents may respond differently once they are more accustomed to consuming breakfast and new PA, and when eating habits have had time to emerge. In addition, we did not assess diet-induced thermogenesis, which is another modifiable component of total energy expenditure and would be expected to negate at least part of the small increase in PAEE of ~258 kJ/d (62 kcal/d) during breakfast omission. Thus, any potential effects of breakfast manipulation on energy balance in our study and subsequent changes in body mass are unlikely to be meaningful [30]. Our drop-out rate and exclusion criteria based on fidelity and complete datasets also meant that our final sample was lower than expected for the dietary and VAS analyses (i.e., our secondary outcome variables), which may have affected our findings and are common issues when working with this population. Additionally, individual chronotype, circadian rhythms, PA habits, breakfast habits, and menstrual cycle phase could all influence the nature of the relationship between breakfast, PA, and diet, which we did not assess and may limit the generalisability of our conclusions. Furthermore, the potential interaction of our findings with the quantity, composition [22,23,24], and timing of the breakfast meal was beyond the scope of the present study and requires examination to inform an evidence-based ‘definition’ of breakfast. Finally, the increased risk of making a type I error from the testing of multiple secondary outcomes is a limitation of the statistical methods; however, as the sample size estimation was based on our primary outcome, there may been an increased risk of type II error for the secondary outcomes.

In conclusion, PA duration and PAEE were not meaningfully affected when adolescent girls who were habitual breakfast skippers consumed breakfast as compared with when they omitted breakfast over seven consecutive days. Total daily energy intake was almost identical during breakfast consumption and omission, with breakfast omission increasing perceptions of appetite during the early part of the morning and tending to increase carbohydrate intake after the 10:30 breakfast omission cut-off. Nevertheless, consuming breakfast can contribute to higher fibre intakes in ‘breakfast skipping’ adolescent girls. Considering the limited and inconclusive evidence to date, future research that accounts for the potential impact of breakfast habits will be important to understand the causal nature of associations between breakfast consumption and energy balance in adolescent girls.

## Figures and Tables

**Figure 1 nutrients-13-04261-f001:**
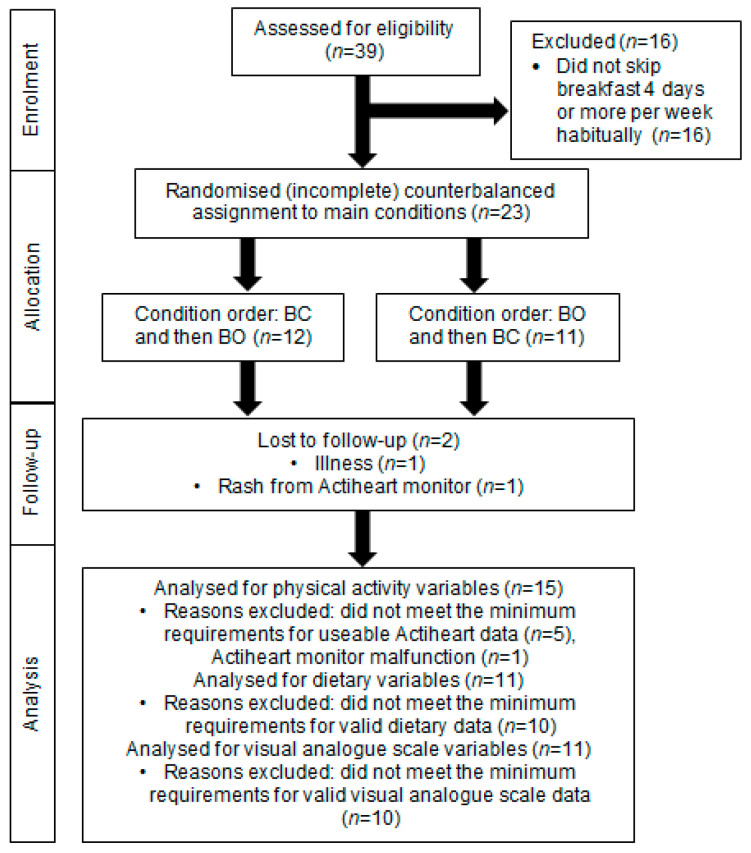
Schematic representation of recruitment, enrolment, and follow-up of adolescent girls who participated in the randomised crossover trial comparing seven days of daily breakfast consumption (BC) with seven days of breakfast omission (BO). BC was the consumption of a standardised breakfast with an energy content equivalent to 25% of individual resting metabolic rate before 09:00 for seven consecutive days; BO was the abstinence from all energy-providing nutrients until at least 10:30 for seven consecutive days.

**Figure 2 nutrients-13-04261-f002:**
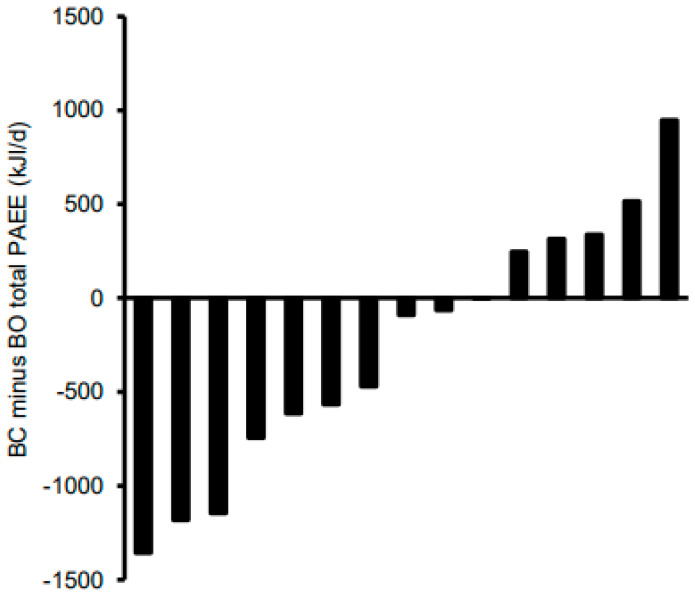
Individual total daily physical activity energy expenditure (PAEE) responses of adolescent girls who participated in the randomised crossover trial comparing seven days of breakfast consumption (BC) with seven days of breakfast omission (BO). BC was the consumption of a standardised breakfast with an energy content equivalent to 25% of individual resting metabolic rate before 09:00 for seven consecutive days; BO was the abstinence from all energy-providing nutrients until at least 10:30 for seven consecutive days.

**Figure 3 nutrients-13-04261-f003:**
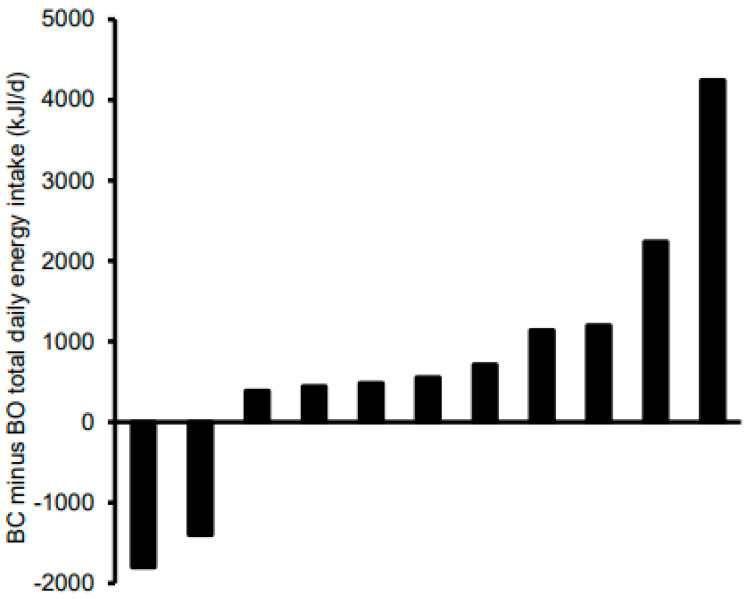
Individual total daily energy intakes of adolescent girls who participated in the randomised crossover trial comparing seven days of breakfast consumption (BC) with seven days of breakfast omission (BO). BC was the consumption of a standardised breakfast with an energy content equivalent to 25% of individual resting metabolic rate before 09:00 for seven consecutive days; BO was the abstinence from all energy-providing nutrients until at least 10:30 for seven consecutive days.

**Figure 4 nutrients-13-04261-f004:**
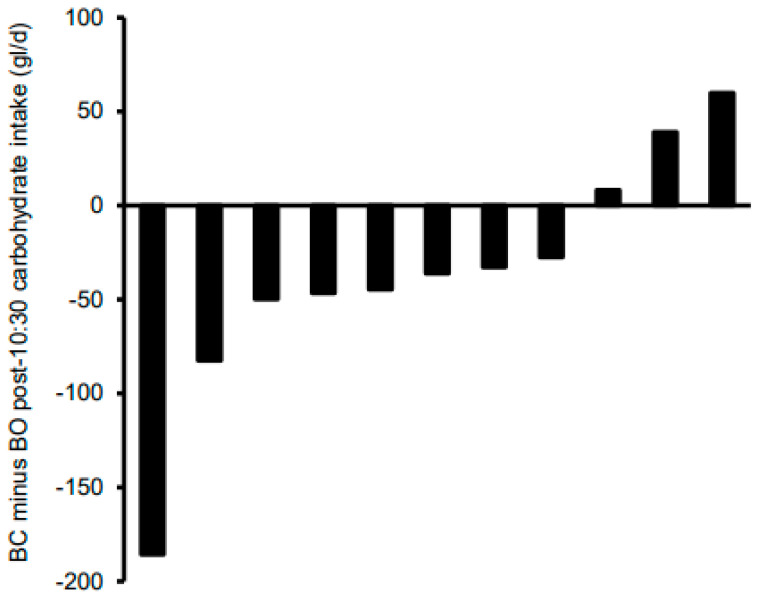
Individual post-10:30 carbohydrate intakes of adolescent girls who participated in the randomised crossover trial comparing seven days of breakfast consumption (BC) with seven days of breakfast omission (BO). BC was the consumption of a standardised breakfast with an energy content equivalent to 25% of individual resting metabolic rate before 09:00 for seven consecutive days; BO was the abstinence from all energy-providing nutrients until at least 10:30 for seven consecutive days.

**Table 1 nutrients-13-04261-t001:** Characteristics of adolescent girls who participated in the randomised crossover trial comparing seven days of daily breakfast consumption (BC) with seven days of breakfast omission (BO) ^1^.

	PA Analyses Sample (*n* = 15)	Diet Analyses Sample (*n* = 11)	VAS Analyses Sample (*n* = 11)
Age (y)	13.3 ± 0.7	13.3 ± 0.8	13.3 ± 0.7
Stature (m)	1.58 ± 0.06	1.57 ± 0.06	1.60 ± 0.06
Body mass (kg)	52.9 ± 7.6	53.3 ± 11.0	54.5 ± 10.1
Body fat %	26.7 ± 5.5	27.0 ± 5.8	26.7 ± 5.9
Waist circumference (cm)	69.7 ± 8.5	68.3 ± 8.9	69.5 ± 9.1
BMI (kg∙m^−2^)	21.3 ± 3.2	21.6 ± 3.8	21.3 ± 3.8
BMI classification (*n* NO, OW, OB) ^2^	11, 3, 1	8, 2, 1	8, 2, 1
Breast development (stage) ^3^	4 (0)	4 (0)	4 (0)
Pubic hair (stage) ^3^	4 (0)	4 (0)	4 (0)
RMR (kJ/d)	6325 ± 1195	6172 ± 1420	6535 ± 1245
Weekdays skip breakfast habitually (d/week) ^4^	4 ± 1	4 ± 1	4 ± 1
Weekend days skip breakfast habitually (d/week) ^4^	1 ± 1	1 ± 0	1 ± 1
Weekly days skip breakfast habitually (d/week) ^4^	5 ± 1	5 ± 1	5 ± 1
Habitual weekday cereal-based breakfast consumption (*n*)	5	3	4
Habitual weekend cereal-based breakfast consumption (*n*)	5	4	5
Habitual weekday breakfast consumption time (h:min)	08:16 ± 01:08	08:09 ± 00:51	08:29 ± 01:25
Habitual weekend breakfast consumption time (h:min)	09:51 ± 01:40	09:37 ± 01:36	09:15 ± 01:50

^1.^ Values are mean ± SDs or medians (IQRs). BC was the consumption of a standardised breakfast for seven consecutive days; BO was the abstinence from all energy-providing nutrients until at least 10:30 for seven consecutive days. PA, physical activity; VAS, visual analogue scale; NO, non-overweight; OW, overweight; OB, obese; RMR, resting metabolic rate. ^2.^ BMI classification according to the International Obesity Task Force [33]. ^3.^ Five stages of breast and pubic hair development described by Tanner [36].^4.^ Less than 50 kcal consumed before 10:30.

**Table 2 nutrients-13-04261-t002:** Nutrient content of the breakfast providing 25% of individual resting metabolic rate to adolescent girls who participated in the randomised crossover trial that compared seven days of daily breakfast consumption (BC) with seven days of breakfast omission (BO) ^1^.

	PA Analyses Sample (*n* = 15)	Dietary Analyses Sample (*n* = 11)	VAS Analyses Sample (*n* = 11)
Energy (kJ)	1578 ± 303	1543 ± 355	1634 ± 311
Carbohydrate (g)	68.5 ± 18.4	63.7 ± 16.5	72.5 ± 19.9
Fat (g)	6.4 ± 4.0	4.9 ± 1.0	6.8 ± 4.5
Protein (g)	13.3 ± 2.6	13.2 ± 2.9	13.8 ± 2.8
Fibre (g)	8.1 ± 4.7	9.3 ± 5.0	7.9 ± 4.6

^1^ Values are mean ± SDs. BC was the consumption of a standardised breakfast for seven consecutive days; BO was the abstinence from all energy-providing nutrients until at least 10:30 for seven consecutive days. PA, physical activity; VAS, visual analogue scale.

**Table 3 nutrients-13-04261-t003:** Free-living physical activity energy expenditure and duration of adolescent girls who participated in the randomised crossover trial comparing seven days of breakfast consumption (BC) with seven days of breakfast omission (BO) ^1^.

	BC			BO			95% CI for BC vs. BO ^2^
	Wake–10:30	10:30–15:30	15:30–bed	Wake–10:30	10:30–15:30	15:30–bed	
**PAEE (kJ/d)**							
Sedentary	54 (40–72)	72 (60–85)	60 (45––78)	54 (41–72)	73 (61–87)	65 (48–87)	−12–5%
Light	321 (233–442)	342 (286–409)	260 (194–349)	321 (238–433)	369 (309–441)	264 (191–365)	−12–7%
Moderate	168 (84–338)	228 (142–366)	152 (79–292)	192 (99–372)	257 (160–411)	148 (73–300)	−24–13%
Vigorous	12 (5–35)	23 (11–47)	16 (6–41)	17 (6–44)	37 (18–76)	15 (5–41)	−40–4%
Total	601 (417–868)	712 (567–894)	547 (389–770)	666 (471–942)	857 (682–1076)	550 (379–797)	−18–1%
**PA Duration (min/d)**							
Sedentary	202 (170–241)	189 (172–207)	209 (178–245)	196 (167–231)	170 (155–187)	205 (172–245)	0–11%
Light	67 (49–90)	74 (64–86)	59 (45–78)	67 (50–89)	81 (70–95)	62 (46–84)	−13–5%
Moderate	15 (7–30)	19 (12–32)	13 (7–25)	17 (9–32)	22 (13–35)	13 (6–26)	−23–12%
Vigorous	1.44 (0.79–2.63)	2.14 (1.37–3.34)	1.93 (1.09–3.42)	1.73 (0.97–3.08)	2.95 (1.89–4.61)	1.78 (0.97–3.27)	−25–2%

^1.^ Based on natural log-transformed data with values presented as a geometric mean (95% confidence interval (CI)) adjusted for minutes of useable data, *n* = 15. ^2.^ A total of 95% CI for the percentage difference of the geometric means between the experimental conditions using a condition by time-of-day linear mixed model adjusted for minutes of useable data and condition order. BC was the consumption of a standardised breakfast with an energy content equivalent to 25% of individual resting metabolic rate before 09:00 for seven consecutive days; BO was the abstinence from all energy-providing nutrients until at least 10:30 for seven consecutive days. PAEE, physical activity energy expenditure; PA, physical activity.

**Table 4 nutrients-13-04261-t004:** Dietary intakes of adolescent girls who participated in randomised crossover trial comparing seven days of breakfast consumption (BC) with seven days of breakfast omission (BO) ^1^.

	BC			BO			95% CI for Total in BC vs. BO ^2^	95% CI for Post-10:30 in BC vs. BO ^2^
	Total	10:30–15:30	15:30–Bed	Total	10:30–15:30	15:30–Bed		
Energy intake (kJ/d)	4206 (3157–5604)	1726 (1041–2863)	1181 (712–1959)	4078 (3063–5433)	2026 (1222–3360)	1628 (982–2700)	−19–31%	−50–23%
CHO (g/d)	118 (82–169)	42 (24–75)	37 (20–67)	127 (88–182)	69 (39–122)	46 (26–82)	−32–28%	−54–7%
Fat (g/d)	35 (25–49)	16 (9–28)	12 (7–21)	38 (27–53)	17 (10–29)	16 (9–27)	−35–33%	−46–38%
Protein (g/d)	42 (29–59)	19 (12–32)	13 (8–21)	32 (22–45)	13 (8–22)	15 (9–25)	−5–83%	−27–67%
Fibre (g/d)	10.0 (7.1–14.2)	2.1 (1.0–4.3)	1.1 (0.5–2.3)	4.7 (3.3–6.7	2.7 (1.4–5.1)	1.4 (0.7–2.7)	34–236%	−61–55%

^1.^ Based on natural log-transformed data, with values presented as a geometric mean (95% confidence interval (CI)), *n* = 11. ^2.^ 95% CI for the percentage difference of geometric means between the experimental conditions using a between-condition (for total intakes) or condition by time-of-day (for post-10:30 intakes) linear mixed model adjusted for condition order; values in bold are significant. BC was the consumption of a standardised breakfast with an energy content equivalent to 25% of individual resting metabolic rate before 09:00 for seven consecutive days; BO was the abstinence from all energy-providing nutrients until at least 10:30 for seven consecutive days. CHO, carbohydrate.

**Table 5 nutrients-13-04261-t005:** Perceived appetite, tiredness, and energy levels (measured in mm out of 100) of adolescent girls who participated in the randomised crossover trial comparing seven days of breakfast consumption (BC) with seven days of breakfast omission (BO) ^1^.

	BC			BO			95% CI for BC vs. BO ^2^
	Waking	09:00	10:30	Waking	09:00	10:30	
Hunger	35 (24–47)	30 (18–41)	34 (22–46)	24 (12–35)	41 (29–52)	54 (42–66)	−13.3–0.4
Fullness	46 (31–61)	56 (41–70)	53 (38–67)	54 (40–69)	43 (29–58)	33 (18–47)	0.2–15.9
Tiredness	48 (36–59)	38 (26–49)	34 (22–45)	46 (35–58)	36 (25–48)	33 (22–45)	−4.3–6.2
Energy	38 (30–46)	48 (40–56)	60 (52–68)	37 (29–45)	50 (42–58)	56 (48–64)	−4.5–6.0

^1.^ Values are estimated marginal mean (95% CIs), *n* = 11. ^2.^ A total of 95% confidence interval (CI) of the mean absolute difference between the experimental conditions; values in bold are significant. BC was the consumption of a standardised breakfast with an energy content equivalent to 25% of individual resting metabolic rate before 09:00 for seven consecutive days; BC was the abstinence from all energy-providing nutrients until at least 10:30 for seven consecutive days.

## Data Availability

The data presented in this study are available on request from the corresponding author.

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
