# Peer review of "Breakfast Consumption Suppresses Appetite but Does Not Increase Daily Energy Intake or Physical Activity Energy Expenditure When Compared with Breakfast Omission in Adolescent Girls Who Habitually Skip Breakfast: A 7-Day Randomised Crossover Trial"

_nutrients, 2021, doi:10.3390/nu13124261_

Round 1
Reviewer 1 Report
Authors of the manuscript entitled "Breakfast consumption suppresses appetite but does not increase daily energy intake or physical activity energy expenditure when compared with breakfast omission in adolescent girls who habitually skip breakfast: a 7-day randomised cross-over trial" present very interesting results of the study. In my opinion this manuscript was very well prepared. The presented study was properly prepared and conducted therefore, I believe the manuscript may be published. Skipping breakfast is a significant problem that has recently been observed mainly in children and adolescents therefore, this research undertaken by Authors is of high scientific importance.
Author Response
We thank the Review for their very positive, supportive comments regarding our manuscript. Although the recommendation that the manuscript should be accepted for publication with no revisions required meant that we have not made revisions based on this review, some minor revisions have been made in light of the comments from Reviewer 2.
Reviewer 2 Report
The reading of the paper suggests some comments:
- The research question posed by the authors is very interesting and the design of the study is well organized and described in detail.
- Materials and Methods Section. 1) Revise Flow-chart Fig.1: N. of subjects in follow-up phase was 19, while in the analysis phase resulted in 21, clarify. 2) In Table 1, also consider BMI classification according to IOTF (normal-weight versus overweight or obese). 3) Because it is difficult to truly assess “free-living” physical activity, if possible, add information about planned session activity (for example, extra-curricular activities, or physical activity at school) for each participant in each group.
Author Response
We thank the Review for their positive, supportive comments regarding our manuscript and for the opportunity to revise and improve our original version. We hope that the concerns raised within the Materials and Methods, Results and Discussion sections have been addressed sufficiently within our point-by-point rebuttal below and within the tracked changes in the manuscript.
1) Revise Flow-chart Fig.1: N. of subjects in follow-up phase was 19, while in the analysis phase resulted in 21, clarify.
Response: To clarify, 23 participants commenced the study, 2 were lost to follow up, leaving 21 who completed the study. After that, those included in analyses depended on the outcome variable, as follows: 15 were analysed for physical activity variables (6 were excluded for the following reasons: did not meet the minimum requirements for useable Actiheart data (n=5), Actiheart monitor malfunction (n=1)), 11 were analysed for dietary variables (10 were excluded for not meeting the minimum requirements for valid dietary data) and 11 were analysed for visual analogue scale variables (10 were excluded for not meeting the minimum requirements for valid visual analogue scale data). We hope that this clarifies any confusion with the flow of the participants outlined in Figure 1.
2) In Table 1, also consider BMI classification according to IOTF (normal-weight versus overweight or obese).
Response: We thank the Reviewer for their suggestion and have made the suggested amendment to Table 1. In addition, we have added details of the weight status classification to the Methods (see section 2.3 Preliminary measurements).
3) Because it is difficult to truly assess “free-living” physical activity, if possible, add information about planned session activity (for example, extra-curricular activities, or physical activity at school) for each participant in each group.
Response: We agree that ‘free-living physical activity’ is very challenging to truly assess and that this requires acknowledgement within the manuscript. As such, as stated in the Methods, ‘the experimental conditions did not coincide with anticipated changes in PA or dietary habits (e.g., holidays, school sports days), as confirmed with the participants, their parents and teachers’. Also, the Discussion states ‘On this note, environmental constraints that are inherent to the lives of adolescent girls, including the school timetable, extra-curricular activities, family routines and commitments, mean that it is difficult to truly assess ‘free-living’ PA responses in this population that would be amenable to any intervention effects – thus, this may have contributed to the lack of difference in PA between the conditions.’ Unfortunately, we did not collect data about planned session activity (for example, extra-curricular activities, or physical activity at school) for each participant in each group. Yet, we agree this would be a useful suggestion for future related research; thus, we have now added ‘and future research may consider collecting data on such planned activities’ to our acknowledgement of this issue within the Discussion (please see the tracked changes within our revised manuscript). We hope that this addresses the issues raised from the Reviewer sufficiently.